# Influence of Magnesium Source on the Mechanochemical Synthesis of Magnesium-Substituted Hydroxyapatite

**DOI:** 10.3390/ma17020416

**Published:** 2024-01-14

**Authors:** Natalia V. Bulina, Natalya V. Eremina, Svetlana V. Makarova, Irina A. Borodulina, Olga B. Vinokurova, Leon A. Avakyan, Ekaterina V. Paramonova, Vladimir S. Bystrov, Olga A. Logutenko

**Affiliations:** 1Institute of Solid State Chemistry and Mechanochemistry, Siberian Branch of the Russian Academy of Sciences, Kutateladze Str. 18, 630090 Novosibirsk, Russia; eremina@solid.nsc.ru (N.V.E.); makarova@solid.nsc.ru (S.V.M.); ir.an.bor@yandex.ru (I.A.B.); olga_logutenko@mail.ru (O.A.L.); 2Physics Faculty, Southern Federal University, 344090 Rostov-on-Don, Russia; laavakyan@sfedu.ru; 3Institute of Mathematical Problems of Biology—Branch of Keldysh Institute of Applied Mathematics, Russian Academy of Sciences, 142290 Pushchino, Russia; ekatp11@gmail.com (E.V.P.); vsbys@mail.ru (V.S.B.)

**Keywords:** density functional theory, doping, hydroxyapatite, magnesium, mechanochemistry, substitution, synthesis, thermal stability

## Abstract

Magnesium, as one of the most abundant cations in the human body, plays an important role in both physiological and pathological processes. In this study, it was shown that a promising biomedical material, Mg-substituted hydroxyapatite (Mg-HA), can be synthesized via a fast mechanochemical method. For this method, the nature of magnesium-containing carriers was shown to be important. When using magnesium oxide as a source of magnesium, the partial insertion of magnesium cations into the apatite structure occurs. In contrast, when magnesium hydroxide or monomagnesium phosphate is used, single-phase Mg-HA is formed. Both experimental and theoretical investigations showed that an increase in the Mg content leads to a decrease in the lattice parameters and unit cell volume of Mg-HA. Density functional theory calculations showed the high sensitivity of the lattice parameters to the crystallographic position of the calcium site substituted by magnesium. It was shown experimentally that the insertion of magnesium cations decreases the thermal stability of hydroxyapatite. The thermal decomposition of Mg-HA leads to the formation of a mixture of stoichiometric HA, magnesium oxide, and Mg-substituted tricalcium phosphate phases.

## 1. Introduction

Hydroxyapatite (HA), Са_10_(РО_4_)_6_(ОН)_2_, is widely used in various fields of medicine, including traumatology and orthopedics, craniofacial surgery and dental technology, therapy, and cosmetology [1]. HA has become a material for the preparation of biocompatible ceramic products, composites, bone defect fillers, medical cements, and implants, as well as a drug carrier for targeted drug delivery [1,2,3]. This wide medical application is due to its similarity to the minerals of human bone and dental tissue.

The HA unit cell contains ten calcium cations, six PO_4_^4−^ tetrahedrons, and two hydroxyl groups located on the *c*-axis (Figure 1). The hydroxyl groups are surrounded by calcium cations forming through-channels in the HA crystal [4]. Stoichiometric HA has a Ca/P atomic ratio of 1.67. Its structure is remarkable since all ions in the HA structure can be substituted, and replacement with both isovalent and heterovalent ions of other chemical elements or their chemical groups is possible [5,6]. 

Magnesium is the fourth most abundant cation in the human body after calcium, potassium, and sodium [7]. Magnesium is present in various human organs and plays an important role in a variety of processes. Approximately 60% of the magnesium is located in bone tissue, and it is almost 1% of the total bone mineral content [7]. Magnesium ion deficiency can directly affect bone properties by altering the structure and size of bone crystals of the biogenic HA, as it affects bone metabolism [8,9,10]. Also, magnesium cations are particularly important for bone density. Thus, some studies show that magnesium deficiency promotes osteoporosis, fragility, microfractures of the trabeculae [11,12,13], and the reduction in bone’s mechanical properties [14,15]. Therefore, the substitution of calcium cations with magnesium cations in synthetic HA attracts huge attention [16,17,18]. 

The synthesis of magnesium-substituted HA (Mg-HA) is usually carried out via chemical precipitation from an aqueous solution, which, under certain conditions, results in the formation of a single-phase product [13,16,17,19,20,21,22,23,24,25]. According to Shepherd et al. [26], Mg-HA has the composition Ca_10–x_Mg_x_(PO_4_)_6_(OH)_2_. Ren et al. [19] reported that there is a limit of the magnesium content in HA of *x* = 1.5 when precipitation methods are used. Alternatively, conventional hydrothermal methods [27,28] can be used to obtain single-phase Mg-HA with a higher Mg content limit of *x* = 3 [25,27].

According to previous studies [11,23,25,27,29], the introduction of magnesium ions into the HA crystal lattice leads to a decrease in the cell parameters due to the significantly smaller radius of Mg^2+^ (0.65 Å) compared to Ca^2+^ (0.99 Å). However, Nagyné-Kovács et al. reported a decrease in the unit cell parameter *a* and an increase in the parameter *c* [21], and an increase in the parameter *a* was also reported by Farzadi et al. [22]. In these studies, the same method of Mg-HA synthesis was used, but with different reagents and synthesis stages. It is expected that the synthesis conditions have great significance and affect the composition and properties of as-synthesized HA [30].

Several studies showed that magnesium complicates the crystallization of apatite and destabilizes its structure, which promotes its thermal transformation to β-tricalcium phosphate (β-TCP) in the 700–900 °C temperature range [13,17,23,31,32,33,34]. In contrast, other studies reported that Mg-HA with a low degree of substitution remains stable at temperatures higher than 900 °C [11,17,20,21,29].

The aim of the present work is to explore the possibility of preparing Mg-HA using a simple mechanochemical method using different magnesium sources, because initial reagents play an important role in this case [35]. Such research has not been conducted previously. Also, it is necessary to clarify the issue of the thermal stability of the as-synthesized Mg-HAs.

## 2. Materials and Methods

### 2.1. Sample Preparation

The mechanochemical synthesis of Mg-HA samples with different concentrations of substituent ions was carried out in a planetary ball mill (AGO-2, Russia) in steel drums with 200 g steel balls at a rotation speed of 1800 rpm. Calcium hydroorthophosphate, CaHPO_4_ (analytical grade, Vekton, Saint Petersburg, Russia), freshly calcined calcium oxide, CaO (analytical grade, Vekton, Saint Petersburg, Russia), magnesium oxide, MgO (pure grade, Vekton, Saint Petersburg, Russia), magnesium hydroxide, Mg(OH)_2_ (analytical grade, Vekton, Saint Petersburg, Russia), and monomagnesium phosphate dihydrate, Mg(H_2_PO_4_)_2_·2H_2_O (analytical grade, Vekton, Saint Petersburg, Russia), were used as the initial reagents.

The initial reagents were mixed in ratios according to reactions (1)–(3), while calcium cations were replaced by magnesium cations, keeping the (Ca + Mg)/P ratio constant at 1.67:6CaHPO_4_ + (4–x)CaO + xMgO → Ca_10–x_Mg_x_(PO_4_)_6_(OH)_2_ + 2H_2_O(1)
6CaHPO_4_ + (4–x)CaO + xMg(OH)_2_ → Ca_10–x_Mg_x_(PO_4_)_6_(OH)_2_ + (2+x)H_2_O(2)
(6–2x)CaHPO_4_ + (4+x)CaO + xMg(H_2_PO_4_)_2_·2H_2_O → Ca_10–x_Mg_x_(PO_4_)_6_(OH)_2_ + (2+3x)H_2_O(3)
where *x* = 0, 0.25, 0.5, 1.0, 1.5, or 2.0.

The mechanochemical treatment of the reaction mixture was carried out for 30 min. The lining of the balls and the inner surfaces of the drums was carried out with the reaction mixture. According to atomic absorption analysis, the iron content in the samples after synthesis did not exceed 0.05 wt.%.

The thermal stability of the as-synthesized compounds was studied in a laboratory electrical furnace SNOL 7,2/1100 (Umega, Utena, Lithuania) by heating them in air at temperatures of 500, 600, 700, 800, 900, and 1100 °C for 2 h. The rate of heating and cooling was 5 °C/min.

### 2.2. Sample Characterization

Powder X-ray diffraction (XRD) patterns of the as-synthesized and heated samples were recorded on a D8 Advance powder diffractometer (Bruker, Karlsruhe, Germany) with Bragg–Brentano geometry using CuKα radiation in the range 2θ = 10°–70° using a step size of 0.02° and an accumulated time per step of 35 s. Phase identification was carried out using the PDF-4 database (ICDD, Release 2011). The unit cell parameters, crystallite size, and phase concentrations were determined using the Rietveld method [36] using Topas 4.2 software (Bruker, Karlsruhe, Germany). The fundamental parameter approach was used to account for the instrumental contribution. 

Fourier transform infrared (FTIR) spectra of the powders were recorded on an Infralum FT-801 spectrometer (Simex, Novosibirsk, Russia) in the wavelength range 550–4000 cm^−1^. Pellets made up of a 4 mg sample and 540 mg of KBr were used for the FTIR study.

Simultaneous thermal analysis (STA) experiments were carried out using an STA 449 F1 Jupiter device (Netzsch, Selb, Germany) equipped with a QMS 403C Aeolos mass spectrometer. The measurements were performed under an argon–oxygen mixture (80:20) at a heating rate of 10 °C/min. The analyzed sample with a mass of 30 mg was placed in a corundum crucible. 

### 2.3. Computational Details

The atomic and electronic structures of HA were calculated using the density functional approach, as described in our previous study [37]. In brief, the electronic band structure and formation energies were calculated using the precise hybrid exchange–correlation functional (i.e., HSE06 [38,39] based on the geometries obtained using a less precise semi-local functional (PBE [40])). The simulation cell selected was a 2 × 2 × 2 orthorhombic supercell (352 atoms, as illustrated in Figure 2), which is sufficiently large to diminish spurious interactions between defects in neighboring cells.

The computations were performed using the QUANTUM ESPRESSO 7.2 software package (open-source community project) [41], operating within the plane-wave formalism. We used norm-conserved pseudo-potentials to represent the core electron states, and the basis set was limited by cut-off energies of 60 Ryd and 120 Ryd for the exchange–correlation operator used in the hybrid functional. Detailed information about the selection of the calculation parameters is available in [37]. 

The HA structure has two crystallographic positions of Ca cations (Figure 2), and both of them can be substituted by Mg cations. We considered the structural models of both types of substitutions, with the substitution degree *x* (count of substitutions per HA unit cell) varying from 0.1 to 2.0.

Previous studies [37] have shown a decrease in the parameters and volume of the HA cell when calcium is substituted for magnesium in the Ca1 and Ca2 positions with the increase in the magnesium concentration *x*, which is in agreement with the results of other authors [19,42]. However, at higher concentrations (*x*~2), the convergence of the calculation was not quite sufficient because of the more substantial structural changes, which required more thorough calculations. In this work, we present the results of such calculations performed using large-scale computational facilities, which allowed us to achieve the necessary predetermined criteria for ion relaxation in the Mg-HA structure and to improve the results presented in [37]. The structural optimization was continued until the difference in total energy in the last two steps remained higher than the predetermined threshold of 10^−4^ eV and the maximum force acting on atoms was above 0.01 eV/Å.

The formation energy E_f_ of calcium-to-magnesium substitution for different values of *n* (the number of magnesium ions in the supercell) was calculated using the following formula [37]:E_f_ = E_tot_ − E_HA_ − n·[μ(Mg) − μ(Ca)],(4)
where E_HA_ is the total energy of the original unsubstituted HA structure measured for a 2 × 2 × 2 × 2 = 8 supercell, which, after full density functional theory (DFT) calculations with the HSE functional, has the value E_HA_ = −180,083.638 eV; E_tot_ is the total energy of the substituted Mg-HA and is calculated after the full relaxation of the supercell with a given number *n* of substitutions; μ(Mg) and μ(Ca) are the chemical potentials of Mg and Ca ions calculated for the reference phases of *hcp* Mg and *fcc* Ca, respectively: μ(Mg) = −1478.734 eV and μ(Ca) = −1003.756 eV. 

The E_f_ values obtained as a result of the calculation are normed per one formula unit (f.u.) of the HA unit cell.

## 3. Results and Discussion

### 3.1. Mechanochemical Synthesis of Mg-HA

Figure 3 shows the XRD patterns of the products prepared from different magnesium-based reagents according to reactions (1)–(3). As can be seen, all the XRD patterns of the samples produced from magnesium oxide have reflections of the initial MgO at 2θ = 43.03° and HA phases (Figure 3a). At *x* = 2, there is also a reflex at 2θ = 31.25°, which is ascribed to the β-Ca_3_(PO_4_)_2_ (TCP) phase. The XRD patterns of the samples synthesized using Mg(OH)_2_ and Mg(H_2_PO_4_)_2_·2H_2_O contain only apatite phase reflections (Figure 3b,c). This leads to the conclusion that reactions (2) and (3) proceed entirely to completion: i.e., all the reagents participate in the formation of the Mg-HA structure, which is not the case for reaction (1). Table 1 shows that the concentration of MgO in the resultant products is about two times less than that introduced into the initial mixture. Therefore, only a part of magnesium oxide reacts with the other components to form Mg-HA via reaction (1). 

The changes in the cell and volume parameters in the substituted samples depending on the concentration of the introduced magnesium oxide confirm the presence of magnesium cations in the structure of as-synthesized Mg-HA (Figure 4a–c). As can be seen, the most significant decrease in these values is observed for the samples prepared via reaction (2), with magnesium hydroxide used as the Mg source. This can be attributed to the different ionic radii of calcium and magnesium cations: R(Ca^2+^) = 1.00 Å and R(Mg^2+^) = 0.72 Å [43]. In the case of MgO, the decrease in these parameters is not as marked because not all the magnesium has reacted, as mentioned above. In the case of monomagnesium phosphate, the larger changes in the parameters at large values of x, unlike the case of magnesium hydroxide, can be explained by the influence of a large number of water molecules released during the chemical interaction of this mixture of the reactants (reaction (3)), which can increase the lattice parameter values. 

An interesting fact is that the concentration of the released water does not affect the crystallite size (Figure 4d). At the same time, the dopant concentration has approximately the same effect on crystallinity for all synthesis variants; the larger the *x*, the smaller the average crystallite size. An essential decrease in the reflection intensity (Figure 3) indicates the substitution limit in the case of the mechanochemical method and other synthesis methods [19,25,27], as well. 

The low reactivity of magnesium oxide in mechanochemical synthesis may be due to its high mechanical strength and strong abrasive resistance (the Mohs hardness value is 5.5 for periclase and 3.5 for calcium oxide), unlike the other reactants in the reaction mixture that are prone to hydration. The authors of [44] reported that, in the first few minutes of the mechanochemical treatment of the initial mixture in a planetary ball mill, the grinding of the reagents occurs, followed by the neutralization reaction with the formation of HA crystallites. The different degrees of grinding of the reagents, resulting from their different hardness values, do not allow the particles to be efficiently distributed throughout the volume of the reaction mixture. Therefore, only a small part of the surface layer of MgO particles is involved in reaction (1). In this case, magnesium-deficient apatite with a low (Са + Mg)/P ratio (<1.67) is formed. With the deficiency of calcium ions in the HA cationic sublattice, the so-called “acidic” apatite Са_10–у_(НРО_4_)_у_(РО_4_)_6–у_(ОН)_2–у_ is formed during mechanochemical synthesis [45]. In our case, since a part of magnesium can be incorporated into the apatite crystal structure, the possible composition of the as-synthesized apatite is Са_10–x_Mg_x–y_(НРО_4_)_y_(РО_4_)_6–у_(ОН)_2–y_, where y < x. Consequently, reaction (1) can be re-written as follows: 6CaHPO_4_ + (4–x)CaO + xMgO → Ca_10–x_Mg_x–y_(HPO_4_)_y_(PO_4_)_6–y_(OH)_2–y_ + yMgO + 2H_2_O(5)

The presence of (HPO_4_)^2−^ groups in the structure of Mg-HA prepared from magnesium oxide is confirmed by the FTIR spectra. As can be seen in Figure 5b,c, at first glance, the spectra of the samples synthesized using different magnesium sources are identical. The absorption bands of the phosphate anion (572, 602, 960, 1048, 1090 cm^−1^) and the bands of the hydroxyl group (630, 3572 cm^−1^) in the HA structure [4] are present in both cases. With the increase in magnesium concentration, the phosphate bands become wider, while the bands of the hydroxyl group disappear. However, upon closer examination of the low-intensity absorption bands of the carbonate group in the HA structure [4], one can notice that in the spectra of the samples synthesized by introducing magnesium oxide, the bands at 1420 and 1485 cm^−1^ (Figure 5a) are almost invisible, unlike the samples prepared from monomagnesium phosphate (Figure 5c). Figure 5b shows that the shape of the additional low-intensity carbonate band at 870 cm^−1^ [4] changes as the magnesium oxide concentration increases; it becomes significantly wider at a high magnesium concentration. When using monomagnesium phosphate, there is only a shift in the carbonate band at 870 cm^−1^ at a high x value (Figure 5d). The shift of this band may be related to the influence of magnesium cations located near the carbonate anion. The broadening of this line observed when using magnesium oxide as the initial reagent cannot be explained by changes in the environment of the carbonate group because there are no main bands at 1420 and 1485 cm^−1^ in the FTIR spectra (Figure 5a). However, there is a new wide band at ~870 cm^−1^, which can be attributed to the (HPO_4_)^2−^ group in calcium-deficient HA [45]. Our previous study showed [45] that the bands of the carbonate and (HPO_4_)^2–^ groups have approximately the same positions but different widths. The broad bands at 1640 and 3430 cm^−1^ are attributed to adsorbed water [4].

The STA data confirm the presence of adsorbed and lattice water in the synthesized samples, which is released in the temperature range of 50–600 °C (Figure 6). The amount of water in the samples is different and consistent with the stoichiometry of reaction Equations (1)–(3). When MgO, Mg(OH)_2_, and Mg(H_2_PO_4_)_2_·2H_2_O are used as the Mg sources, the amount of water released from the samples is 4.1, 7.1, and 7.2 wt.%, respectively. The endoeffects observed in the DTA curves at higher temperatures are obviously related to the decomposition of the substituted apatite structure.

The data shown in Table 2 indicate that the removal of adsorbed water does not change the phase composition of the samples synthesized with the addition of magnesium hydroxide and monomagnesium phosphate. They remain single-phase, in contrast to the samples synthesized with the incorporation of magnesium oxide. Heating the samples obtained with MgO led to an increase in the MgO concentration and the formation of the TCP phase at *x* = 1.5.

Figure 7 shows the changes in the unit cell parameters of the Mg-HA phase in the samples after heating at 500 °C. Comparing the data shown in this figure with those shown in Figure 4, it is evident that, after heating, the parameters keep changing with increasing *x*, but not that much. At the same time, after the removal of water, the lattice parameters and cell volumes of the samples synthesized using Mg(OH)_2_ and Mg(H_2_PO_4_)_2_·2H_2_O are nearly the same and keep decreasing with increasing *x*. This suggests that samples with the same *x* have comparable magnesium concentrations in the Mg-HA lattice. As for the samples synthesized using magnesium oxide, the change in the parameter *c* is very similar to that of the samples obtained using both Mg(OH)_2_ and Mg(H_2_PO_4_)_2_·2H_2_O, although the parameter *a* increases with an increase in *x*. The different change trends in the parameter *a* for the samples obtained with MgO may be caused by an increase in the number of vacancies of OH groups in the Mg-HA structure [4], in accordance with Formula (4). The decrease in the parameter *c*, in this case, is due to the increase in the number of substitutions at the calcium positions with both magnesium and the hydrogen ions of the (HPO_4_)^2−^ group, which is also located in the region of the vacant position of the calcium cation. After water has been removed from the Mg-HA samples, the crystallite size of the apatite phase gradually decreases with increasing *x* for all series of samples (Figure 7d).

### 3.2. Lattice Parameters Predicted by DFT

The purpose of this section is to provide a theoretical insight into the incorporation of Mg^2+^ in the HA crystal lattice, since it is not possible to refine the structure of the substituted apatite experimentally in this case due to its small crystallite size and the low electron density of the substituent cation.

Figure 8 shows the calculated cell parameters for models containing different amounts of Ca1 and Ca2 substitutions. As can be seen, the general tendency is a decrease in the unit cell size with the addition of Mg cations. However, differences in the positions of the substituted Ca cations have a significant effect on the cell parameters at high *x*. In particular, at *x* > 1, the cell parameters *a* and *b* decrease noticeably if magnesium is located in the Ca1 position, while the parameter *c* increases only slightly. In the case of the Ca2 substitutions, all parameters decrease but at a slower rate than in the case of Ca1, so the unit cell volumes at the same *x* in both substitutions have comparable values in the whole range of magnesium concentrations, and they decrease almost linearly with the increasing number of substitutions in the cell.

A comparison of the results of modeling (Figure 8) with the experimental data (Figure 7) suggests that, during the synthesis of Mg-HA, the substitution of calcium by magnesium predominantly occurs in the Ca2 position. This is also indicated by the calculated values of the formation energy (E_f_) of the substitutions of calcium for magnesium in the Ca1 and Ca2 positions, as shown in Figure 9.

Figure 9 shows that the dependence of E_f_ on the magnesium concentration in the Ca1 position has a parabolic shape with a minimum at *x* = 1. The E_f_ values for substitutions in the Ca2 position change irregularly as the concentration *x* increases. As can be seen, all the values except for those at *x* = 1 are below the E_f_ values for substitutions in the Ca1 position, i.e., E_f_(Ca1) > E_f_(Ca2). At a concentration *x* = 1, the formation energy values for both Mg substitutions are close, which means that magnesium cations can coexist in the Ca1 and Ca2 positions. 

From the above, we can conclude that for almost all values of *x*, Mg substitution in the Ca2 position is energetically more favorable than that in the Ca1 position; hence, the substitution in the Ca2 position mainly occurs during Mg-HA synthesis.

### 3.3. Thermal Stability of Mg-HA

The thermal stability of the Mg-HA samples synthesized using Mg(H_2_PO_4_)_2_·2H_2_O was investigated. Figure 10 shows the changes in the XRD patterns of Mg-HA samples at different concentrations *x* of magnesium after heat treatment at different temperatures. As can be seen, the substituted samples are less stable than unsubstituted HA. Upon the decomposition of Mg-HA, reflections of β-TCP and MgO appear in the XRD patterns, which was also reported by Moreira et al. [46]. 

The results of the quantitative phase analysis of the XRD patterns by modeling their profiles showed that the decomposition of Mg-HA proceeds in two stages, and the starting temperature of this process depends on the concentration of magnesium in the Mg-HA structure. As can be seen in Table 3, at a concentration *x* = 0.25, the sample remains single-phase up to 700 °C inclusive, whereas, at 800 °C, a nanosized phase of magnesium oxide is formed. At higher values of *x*, the MgO phase appears earlier at 700 °C. A further increase in the temperature up to 800 °C results in the formation of the second impurity phase, β-TCP, whereas the concentration of MgO increases. The amounts of MgO and β-TCP increase as the magnesium concentration in Mg-HA increases and the Ca/P ratio decreases. 

Figure 11 shows that, unlike the samples heated at 500 °C (Figure 7a–c), the unit cell parameters and volume of the HA phase annealed at 900 °C, with the formation of the impurity phases MgO and β-TCP, barely change with the increasing *x* value. This implies that the HA structure after the formation of the impurity phases does not contain magnesium cations. At the same time, the unit cell parameter values of the β-TCP phase are *a* = 10.34 Å and *c* = 37.20 Å, which are much smaller than those reported by Bohner et al. [47]. Consequently, the β-TCP phase contains magnesium cations, which agrees with the data reported in previous studies [13,33].

Taking into account that the decomposition of Mg-HA proceeds in two stages, with the formation of MgO in the first stage and the substituted β-TCP and MgO in the second stage, the decomposition of Mg-HA can be described by reactions (6) and (7):(6)$${Ca}_{10–x}{\mathrm{Mg}}_{x}({\mathrm{PO}}_{4}{)}_{6}(\mathrm{OH}{)}_{2}\;\overset{t^{o}}{\to }\;{Ca}_{10–x}{\mathrm{Mg}}_{x–y}({\mathrm{PO}}_{4}{)}_{6}(\mathrm{OH}{)}_{2–2y}+\mathrm{yMgO}+yH_{2}O,$$where y ≤ 0.5.
(7)$${Ca}_{10–x}{\mathrm{Mg}}_{x–y}({\mathrm{PO}}_{4}{)}_{6}(\mathrm{OH}{)}_{2–2y}\;\overset{t^{o}}{\to }(1–x){Ca}_{10}({\mathrm{PO}}_{4}{)}_{6}(\mathrm{OH}{)}_{2}+2x{Ca}_{3–x+y+z}{\mathrm{Mg}}_{x–y–z}({\mathrm{PO}}_{4}{)}_{2}+\mathrm{zMgO}+nH_{2}O,$$

In the first stage of the decomposition, proceeding by reaction (6), Mg-HA, which contains a low concentration of cationic vacancies and magnesium oxide, is formed. In the second stage of the decomposition, proceeding by reaction (7) at a slightly higher temperature, the number of hydroxyl groups in the apatite structure decreases to less than 1, so the crystal lattice of the substituted apatite becomes unstable and decomposes into substituted β-TCP and MgO. Magnesium substitution for calcium cations in the structure of β-TCP was also reported in previous studies [21,23,33]. 

The significant difference between the ionic radii of calcium and magnesium may be the reason for the decomposition of Mg-HA. In this case, the magnesium cations, which partially occupy the calcium positions, are located at a much larger distance from their nearest neighboring oxygen anions than calcium cations in the same position. The thermal expansion of the crystal lattice upon heating, resulting in a further increase in the ion–ion distances, leads to the destruction of the ionic crystal at the sites of localization of the magnesium cations, followed by the formation of β-TCP and MgO phases. It is noteworthy that, for all concentrations of the introduced magnesium, the behavior of the crystallite size of the substituted apatite phase after cooling and unsubstituted HA (Figure 12), which does not undergo any structural transformations in this temperature range, have the same dynamics. Even at *x* = 2, when the Mg-HA structure is almost completely decomposed (only 6% HA remains), the changes in the crystallite size have the same dynamics. Consequently, the Mg-HA structure decomposes upon heating. 

A significant decrease in the thermal stability of the Mg-HA samples does not allow for studying their mechanical properties, since such a study is carried out on a dense ceramic material. In this regard, computer modeling becomes very relevant. Our previous study showed that Mg substitutions in the Ca2 position decrease the bulk modulus, indicating a decrease in the mechanical strength of Mg-HA [37].

The inability to obtain dense ceramics from Mg-HA without destroying its crystal lattice also limits the possible applications of Mg-HA. This material cannot be used for the production of ceramic implants or metal implant coatings using long-term, high-temperature processing technology. An exception to this may be technologies applying ultra-fast heat treatment to the material, such as the selective laser melting of apatites with a very low degree of substitution [48].

On the other hand, the reduced degree of crystallinity of Mg-HA, due to point defects in the crystal lattice of HA in the form of magnesium cations, indirectly indicates the increased solubility of this material. To confirm this, further research is needed, which we plan to do in the near future. 

## 4. Conclusions

Mg-substituted HA with the composition Ca_10–x_Mg_x_(PO_4_)_6_(OH)_2_ was synthesized using the mechanochemical method. It was shown that magnesium hydroxide or monomagnesium phosphate should be used as the magnesium-bearing reagent. The interaction of these compounds with the other components of the reaction mixture (CaO and CaHPO_4_) during mechanical treatment in a planetary ball mill results in the formation of a single-phase-substituted apatite directly in the mill without further heat treatment. In the case of magnesium oxide, the reaction does not proceed entirely to completion; as a result, a non-stoichiometric, partially substituted apatite containing an impurity phase of the initial MgO is formed.

Mg-substituted HA prepared by mechanochemical synthesis contains lattice and adsorbed water in amounts of no more than 7.2 wt.%, which is removed upon heating at 500 °C. An increase in the Mg content introduced into HA leads to a decrease in the unit cell parameters, its volume, and the crystallite size, as well. The results of DFT calculations performed on the HA supercell model of 352 atoms showed similar dynamics of changes in the unit cell parameters. Moreover, it was found that the dependence of cell parameters on the degree of substitution significantly deviated from linear behavior at large values of *x* for magnesium substitution in the Ca1 position. A comparison of the experimental and calculated cell parameters suggests that magnesium substitutions in both the Ca1 and Ca2 positions can occur, but the Ca2 position demonstrates behavior closer to that observed under the conditions of mechanochemical synthesis. This result is also confirmed by the consideration of the calculated energies of formation of Mg substitutions in the Ca1 and Ca2 positions.

The study of the thermal stability of Mg-substituted HA shows that the introduction of magnesium cations destabilizes the HA structure; therefore, upon the heating of the samples above 600 °C, Mg-HA decomposes. The starting temperature of thermal decomposition depends on the concentration of magnesium in the sample. The more magnesium in the Mg-HA structure, the lower the temperature at which decomposition starts. Thus, at a concentration *x* = 0.25, the samples remain single-phase up to a temperature of 700 °C, while at higher values of *x*, decomposition starts at 700 °C. The decomposition of Mg-HA proceeds in two stages. In the first stage, the MgO phase is formed. A further increase in the temperature of 100 °C results in the formation of another impurity phase, β-TCP. The concentrations of the released MgO and β-TCP phases increase as the magnesium concentration in Mg-HA increases. After the release of magnesium oxide, the remaining HA phase does not contain magnesium cations in its lattice, while the phase of another phosphate, β-TCP, contains magnesium cations.

The reduced thermal stability of the material limits its use in technologies associated with long-term heat treatment at high temperatures. On the other hand, the reduced size of crystallites due to the presence of magnesium cations and water molecules in the HA lattice should increase the solubility of the material, which makes it more promising for use as a bioresorbable component.

## Figures and Tables

**Figure 1 materials-17-00416-f001:**
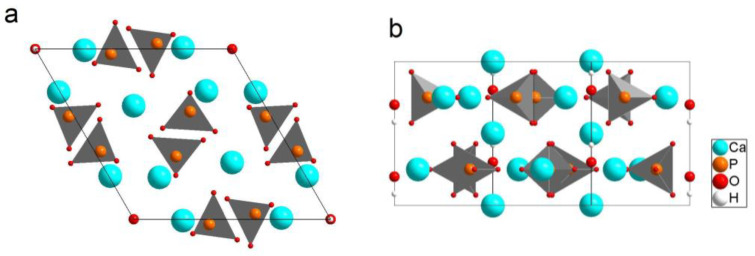
A view of the crystal structure of HA along the *c* axis (**a**) and the *b* axis (**b**).

**Figure 2 materials-17-00416-f002:**
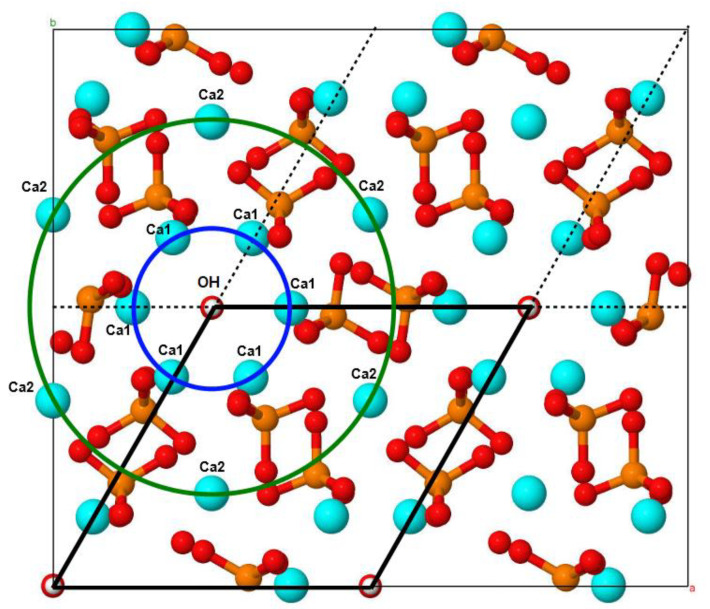
The illustration of the considered HA supercell containing 352 atoms, as viewed from the *c*-axis. The bold black lines illustrate the hexagonal unit cell of HA (contains 44 atoms, Figure 1) in the considered orthorhombic supercell. The blue and green circles show, respectively, the first and second rows of calcium atoms around the OH-channel. Atom color differentiation: cyan—calcium; orange—phosphorus; red—oxygen; white—hydrogen. Adapted from [37].

**Figure 3 materials-17-00416-f003:**
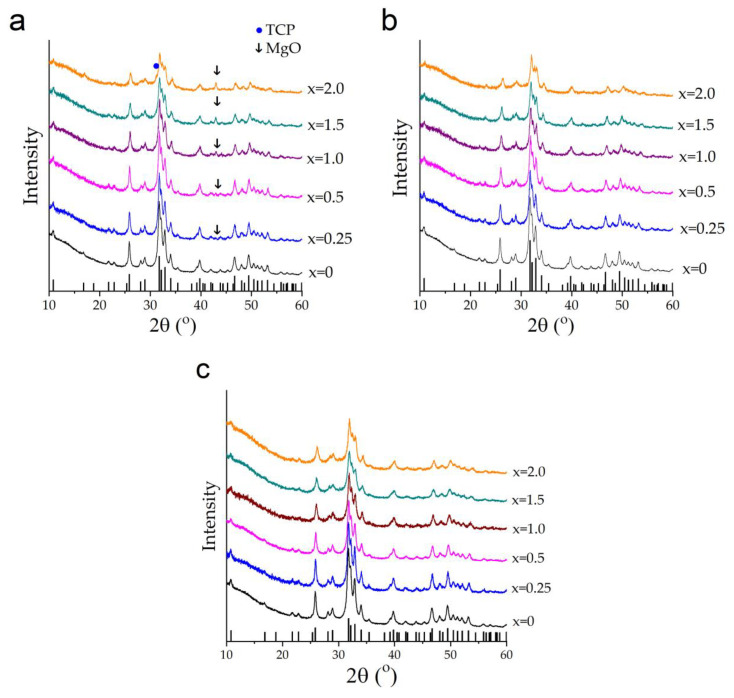
XRD patterns of the samples synthesized with different concentrations of MgO (**a**), Mg(OH)_2_ (**b**), and Mg(H_2_PO_4_)_2_·2H_2_O (**c**). Vertical bars in the (**a**) graph correspond to the HA phase from the ICDD database (PDF No. 40-11-9308).

**Figure 4 materials-17-00416-f004:**
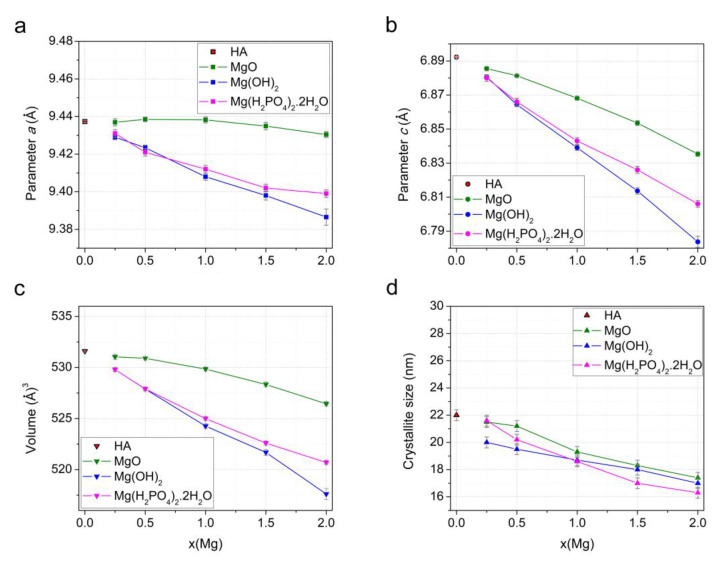
Variation in the parameters *a* (**a**) and *c* (**b**), unit cell volume (**c**), and crystallite size (**d**) of the Mg-HA phase in the as-synthesized samples as a function of magnesium concentration in the initial mixture when using different magnesium-based reagents.

**Figure 5 materials-17-00416-f005:**
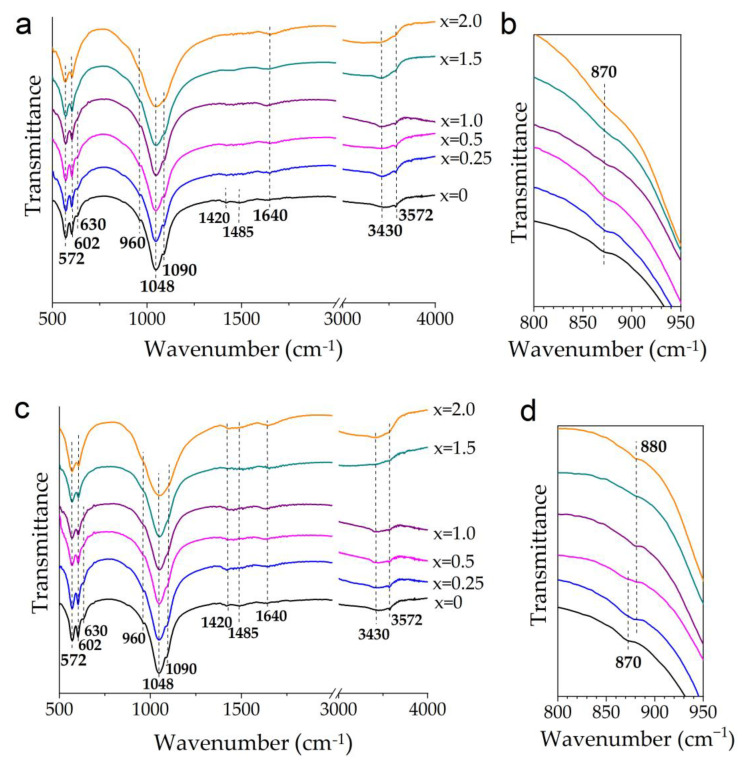
General (**a**,**c**) and enlarged (**b**,**d**) views of FTIR spectra of the samples synthesized with different concentrations of MgO (**a**,**b**) and Mg(H_2_PO_4_)_2_·2H_2_O (**c**,**d**).

**Figure 6 materials-17-00416-f006:**
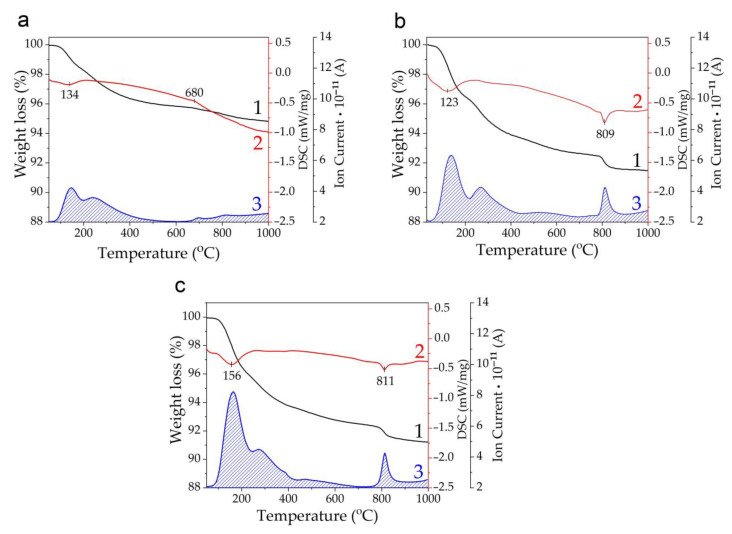
STA of the Mg−HA samples synthesized using MgO (**a**), Mg(OH)_2_ (**b**), and Mg(H_2_PO_4_)_2_·2H_2_O (**c**): 1—weight loss; 2—DSC; 3—water release.

**Figure 7 materials-17-00416-f007:**
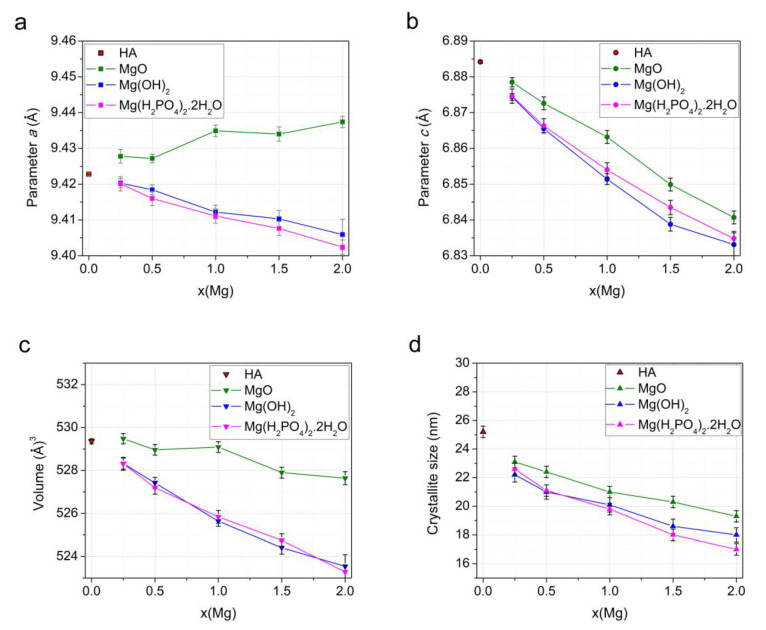
Changes in the parameters *a* (**a**) and *c* (**b**), unit cell volume (**c**), and crystallite size (**d**) of the Mg-HA phase in the samples annealed at 500 °C depending on the magnesium concentration in the initial mixture when using different magnesium-based reagents.

**Figure 8 materials-17-00416-f008:**
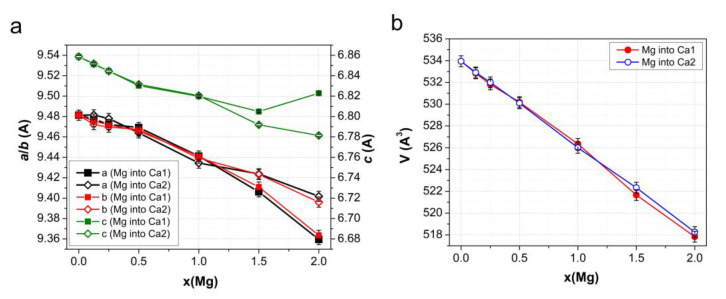
Changes in the parameters (**a**) and volume (**b**) of the Mg-HA unit cell depending on the concentration *x* of magnesium cations in Ca1 and Ca2 positions. The results of DFT calculations for substitution in a supercell consisting of 8 unit cells are shown.

**Figure 9 materials-17-00416-f009:**
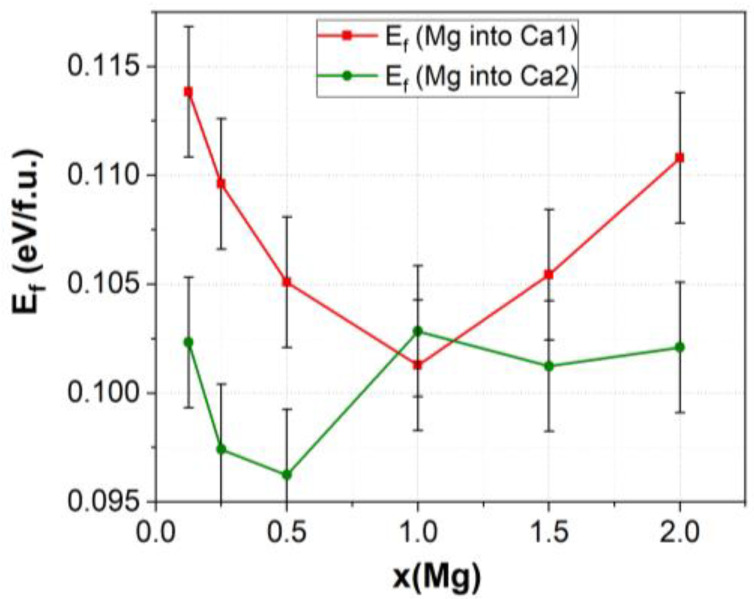
Formation energy E_f_ of calcium substitution by magnesium in the supercell at the Ca1 and Ca2 positions as a function of the number of substituted *x* ions.

**Figure 10 materials-17-00416-f010:**
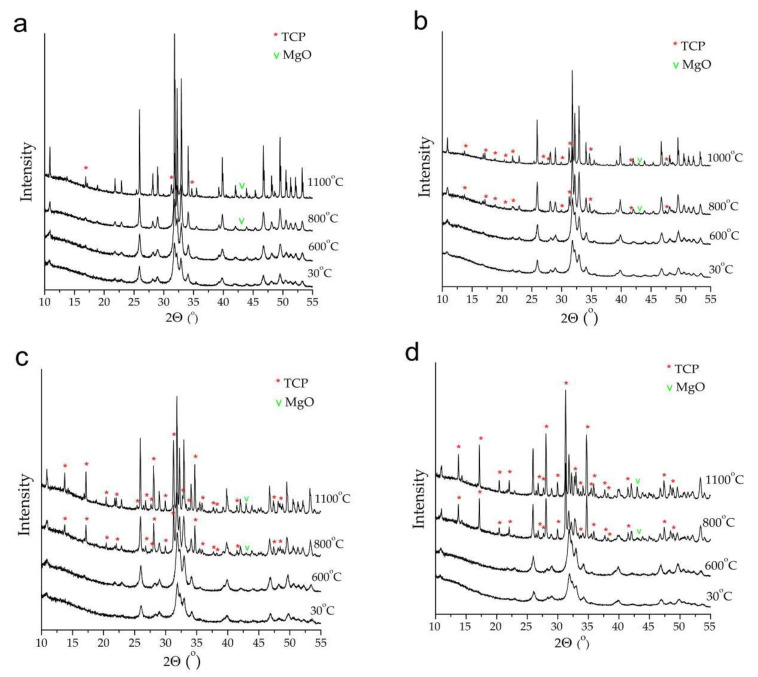
XRD patterns of Mg-HA samples with magnesium concentrations *x* = 0.25 (**a**), *x* = 0.5 (**b**), *x* = 1.0 (**c**), and *x* = 2.0 (**d**) after heat treatment at different temperatures. Unmarked reflections are attributed to the HA phase.

**Figure 11 materials-17-00416-f011:**
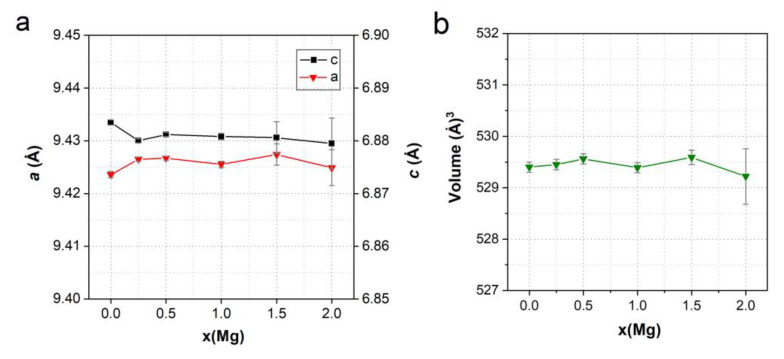
Variation in the parameters *a* and *c* (**a**) and unit cell volume (**b**) of the HA phase in samples with different concentrations of the introduced magnesium after annealing at 900 °C.

**Figure 12 materials-17-00416-f012:**
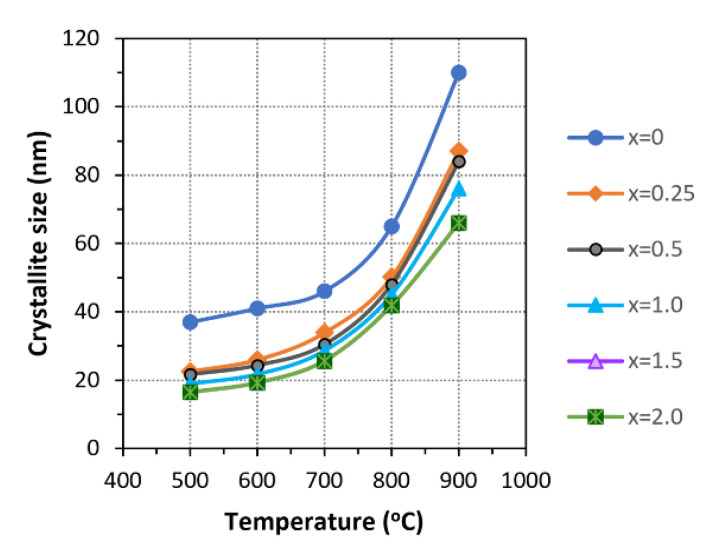
Evolution of the apatite crystallite size with temperature upon heating Mg-HA containing different magnesium concentrations.

**Table 1 materials-17-00416-t001:** Results of the quantitative analysis of the samples produced using magnesium oxide.

Initial Concentration		Powder Composition after Synthesis (wt.%)		
x(Mg)	MgO (wt.%)	Mg-HA	MgO	TCP
0	0	100	0	0
0.25	0.97	99.5	0.5	0
0.5	1.95	99.3	0.7	0
1.0	3.93	98.1	1.9	0
1.5	5.94	96.2	3.8	0
2.0	7.99	82.9	6.4	10.7

**Table 2 materials-17-00416-t002:** Phase composition (wt.%) of the synthesized samples with different *x* after heating at 500 °C.

Initial Concentration of Mg (*x*)	MgO Using (Reaction (1))			Mg(OH)_2_ Using (Reaction (2))	Mg(H_2_PO_4_)_2_·2H_2_O Using (Reaction (3))
	Mg-HA	MgO	TCP	Mg-HA	Mg-HA
0	100	0	0	100	100
0.25	99.2	0.8	0	100	100
0.5	98.6	1.4	0	100	100
1.0	97.5	2.5	0	100	100
1.5	92.6	3.3	4.1	100	100
2.0	74.8	4.1	21.1	100	100

**Table 3 materials-17-00416-t003:** Phase composition (wt.%) of the Mg-HA samples with different magnesium (*x*) concentrations after heating at different temperatures.

*x*(Mg)	Ca/P	Phase	Temperature (°C)					
			600	700	800	900	1100	1200
0.25	1.625	HA	100	100	99.7	94.1	94.1	91.1
		MgO	–	–	0.3	0.5	0.8	0.6
		TCP	–	–	–	5.4	7.2	8.3
0.5	1.583	HA	100	99.9	88.8	81.1	80.7	79.6
		MgO	–	0.1	1.0	1.4	1.3	1.3
		TCP	–	–	10.2	17.5	17.9	19.0
1.0	1.5	HA	100	99.0	70.0	58.0	56.8	54.6
		MgO	–	1.0	2.0	2.6	2.6	2.8
		TCP	–	–	28.0	39.4	40.6	42.6
1.5	1.417	HA	100	96.6	36.2	27.6	28.0	25.7
		MgO	–	3.4	4.7	4.3	4.2	4.1
		TCP	–	–	59.1	68.1	67.8	70.2
2.0	1.333	HA	100	94.3	14.3	6.1	5.1	6.4
		MgO	–	3.1	6.6	5.9	5.7	5.8
		TCP	–	2.6	79.1	88.0	89.3	87.8

## Data Availability

The raw/processed data required to reproduce these results are included in Section 2.

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
