# Peer review of "Influence of Magnesium Source on the Mechanochemical Synthesis of Magnesium-Substituted Hydroxyapatite"

_materials, 2024, doi:10.3390/ma17020416_

Round 1

Reviewer 1 Report

Comments and Suggestions for Authors

The manuscript ” Influence of magnesium source on the mechanochemical synthesis of magnesium-substituted hydroxyapatite” by  Bulina et al. described the preparation of Mg-substituted hydroxyapatite (Mg-HA) using the fast mechanochemical method with different magnesium sources.

The writing lacks clarity and sharpness, and is poorly organized in results and discussion section.  Therefore, I cannot recommend the acceptance of this work in the current form for publication in Materials.

Comments:

1.      Please explain the initial concentration of Mg (x) range from 0~2, why?  

2.      Please show the bone’s mechanical properties of Mg-substituted hydroxyapatite (Mg-HA) with an increase the initial concentration of Mg (x).

3.      Line 156, the significant digits of EHA should be shorted.

4.      The paper should be shortened to become more concise.

Author Response

Firstly, the authors would like to thank the Reviewer for his/her time and consideration.

Our detailed responses on the comments are provided below. The revisions in the manuscript are highlighted in grey.

Question 1: Please explain the initial concentration of Mg (x) range from 0~2, why?  

Response: The aim of the present work was to explore the possibility of Mg-HAs preparation by simple mechanochemical method using different magnesium sources, and to study the structure and thermal stability of the as-synthesized Mg-HAs. The range of the magnesium concentrations was chosen based on the literature data. The results of this study showed that the introduction of magnesium into the crystal lattice of HA reduces crystallinity. At x=2, a significant decrease in the crystallinity is observed, which indicates the existence of a substitution limit for the mechanochemical method of synthesis. In the present study, the limit was not reached, but unfortunately it is not possible to continue synthesis with higher x values at the moment. The discussion concerning the existence of a substitution limit for the mechanochemical method has been added to the manuscript.

Question 2: Please show the bone’s mechanical properties of Mg-substituted hydroxyapatite (Mg-HA) with an increase the initial concentration of Mg (x).

Response: Thank you for this recommendation. Unfortunately, such studies are impossible for this material due to its low thermal stability. Mechanical properties are commonly studied on a ceramic material annealed at high temperature. We have shown that the crystal lattice of Mg-substituted HA is destroyed at ~700 C, so it is impossible to make any ceramic product from this material without destroying the crystal structure of Mg-HA.

Question 3: Line 156, the significant digits of EHA should be shorted.

Response: Thanks for noticing. We have rounded the indicated value.

Question 4: The paper should be shortened to become more concise.

Response: In response to this comment, we critically analyzed the text of our manuscript and came to the conclusion that there is nothing to remove there. The section Result & Discussion contains three short subsections devoted to 1) mechanochemical synthesis; 2) modelling by DFT; 3) thermal stability of the material.  All information provided is important. Moreover, other Reviewers note “logical construction of article” and “good scientific rigor”.

Reviewer 2 Report

Comments and Suggestions for Authors

In this manuscript, the authors have performed very detailed studies on Mg-substituted hydroxyapatite prepared using the mechanochemical method (i.e., ball milling). The significance of this study can be found from the controversy in the literature – for instance, the solubility limit of Mg and the lattice parameter change due to the Mg substitution – as the author discussed in the introduction. Including their theoretical study on the preferential Ca site of Mg, the present study provides comprehensive discussion as well as ground information for potential readers; it is thus worthwhile to be reported. I highly recommend its acceptance, provided the following questions and comments are taken into consideration and addressed:

1. If the authors provide XRD data of the samples before and after ball milling, it would be good to show the effectiveness of ball milling and its success in mechanochemical synthesis. 

2. What does STA stand for? Simultaneous thermal analysis? It needs to be addressed. 

3. In the comparison of the FTIR spectra (Figure 5), the authors claim the presence of (HPO4)2- groups (absorption band at 880 cm-1) only in the sample prepared with MgO. However, in my opinion, it is hard to say that. Any data processing to make this more visible? 

4. Is there any particular reason to use Ar:O (not N or N:O) for the thermal analyses?

5. The authors may consider using one expression to avoid confusion — magnesium dihydrogen phosphate or mono magnesium phosphate. 

Author Response

We appreciate your interest in our study and the meticulous review of our manuscript. Corrections to the manuscript made in accordance with your questions are highlighted in cyan.

Question 1: If the authors provide XRD data of the samples before and after ball milling, it would be good to show the effectiveness of ball milling and its success in mechanochemical synthesis.

Response: Thank you for this recommendation. Yes, comparison of the samples of the initial mixture and the powder after treatment is a good way to show the effectiveness of ball milling. Anyway, from our point of view, Figure 3 also demonstrates the effectiveness of ball milling. Since there are no reflections of impurity phases in the synthesis products (Fig. b and c), this means that the treatment was successful. Therefore, we see no reason to provide XRD data of the samples of the original mixtures (in our case it would be 16 mixtures).

Question 2: What does STA stand for? Simultaneous thermal analysis? It needs to be addressed.

Response: Thanks for the noticing. STA is an abbreviation from Simultaneous thermal analysis. The relevant correction has been made.

Question 3: In the comparison of the FTIR spectra (Figure 5), the authors claim the presence of (HPO4)2- groups (absorption band at 880 cm-1) only in the sample prepared with MgO. However, in my opinion, it is hard to say that. Any data processing to make this more visible? 

Response: Thank you for this comment. We have inserted a magnified view of this area in Figure 5. In addition, we have discussed this figure in more details in the revised manuscript.

Question 4: Is there any particular reason to use Ar:O (not N or N:O) for the thermal analyses?

Response: There are three reasons of using argon instead of nitrogen: 1) argon has higher calorimetric affinity than nitrogen; 2) argon is more convenient to use because its fluidity is less than that of nitrogen; 3) lower cost of gas, which is a consumable item, is an important point.

Question 5: The authors may consider using one expression to avoid confusion — magnesium dihydrogen phosphate or mono magnesium phosphate

Response: Thanks for your comment. The relevant correction has been made.

Reviewer 3 Report

Comments and Suggestions for Authors

The manuscript entitled "Influence of magnesium source on the mechanochemical synthesis of magnesium-substituted hydroxyapatite" describes a rapid mechanochemical method for synthesizing magnesium-substituted hydroxyapatite (Mg-HA).

The study is interesting and was written with good scientific rigor. However, some minor remarks need to be addressed.

Keywords: The keywords should be in alphabetical order, and "KEYWORDS" should not contain the same words that are within the title of the text. Thus, these should be changed appropriately.

Introduction Line 31-33: Add the following reference to support this statement.

M&M: Line 31 - Add software producer and country.

Please add the limitation of the study at the end of the discussion and propose further studies that need to be conducted with this material.

Author Response

The authors would like to thank the reviewer for his time and appreciation of the work.

Our detailed responses are provided below. The changes in the manuscript are highlighted in yellow.

Question 1: The keywords should be in alphabetical order, and "KEYWORDS" should not contain the same words that are within the title of the text. Thus, these should be changed appropriately.

Response: Thanks for the noticing. The corresponding correction has been made.

Question 2: Introduction Line 31-33: Add the following reference to support this statement.

Response: The reference has been added and some corrections have been made in this sentence. Thank you for your comment.

 Question 3: M&M: Line 131 - Add software producer and country. 

Response: This software does not have a specific producer and country respectively. This is a community project and we have added this information in the manuscript.

Question 4: Please add the limitation of the study at the end of the discussion and propose further studies that need to be conducted with this material.

Response: Thanks for the recommendation. We have added the following text:

A significant decrease in the thermal stability of the Mg-HA samples does not allow studying their mechanical properties, since such a study is carried out on a dense ceramic material. In this regard computer modelling becomes very relevant. Our previous study has showed that Mg substitutions in the Ca2 position decrease the bulk modulus, indicating a decrease in the mechanical strength of Mg-HA [37].

The inability to obtain dense ceramics from Mg-HA without destroying its crystal lattice also limits the possible applications of Mg-HA. This material cannot be used for the production of ceramic implants or metal implant coatings using the long-term, high-temperature processing technology. An exception to this may be technologies applying ultra-fast heat treatment of the material, such as selective laser melting of apatites with a very low degree of substitution [48].

On the other hand, the reduced degree of crystallinity of Mg-HA, due to the point defects in the crystal lattice of HA in the form of magnesium cations, indirectly indicates the increased solubility of this material. To confirm this, further research is needed, which we plan to do in the nearest future. 

Reviewer 4 Report

Comments and Suggestions for Authors

Article: Influence of magnesium source on the mechanochemical synthesis of magnesium-substituted hydroxyapatite for publication in Materials is good. The subject is interesting. This work presents Magnesium, as one of the most abundant cations in the human body, plays an important role in both physiological and pathological processes. In this study, it was shown that a promising biomedical material Mg-substituted hydroxyapatite (Mg-HA) can be synthesized via fast mechanochemical method. For this method, the nature of magnesium-contained carriers was shown to be important. When using magnesium oxide as a source of magnesium, the partial insertion of magnesium cations into the apatite structure occurs. In contrast, when magnesium hydroxide or mono magnesium phosphate is used, a single-phase Mg-HA is formed. Both experimental and theoretical investigations showed that an increase in the Mg content leads to a decrease in the lattice parameters and unit cell volume of Mg-HA. Density functional theory calculations showed the high sensitivity of the lattice parameters towards the crystallographic position of the calcium site substituted by magnesium. It was shown experimentally that insertion of magnesium cations decreases the thermal stability of the hydroxyapatite. The thermal decomposition of Mg-HA leads to the formation of a mixture of stoichiometric HA, magnesium oxide and Mg-substituted tricalcium phosphate phasesThe presented article is interesting and a construction of article is logical. The work is relevant and practical. Clarity of expression and communication of ideas, readability and discussion of concepts is good.

However, some corrections are needed:

1.      Figures 4, 5 and 6 should be corrected.

2.      The publication should describe the purity of the reagents used for the synthesis.

3.      It would be good to write what the Ca/P ratio was in individual samples after Mg substitution.

4.      Novelty elements should be better highlighted in the introduction. Papers should be cited in Introduction section; for example:

The influence of the hydroxyapatite synthesis method on the electrochemical, surface and adsorption properties of hydroxyapatite Adsorption Science & Technology   35(5–6) (2017) 507–518

A study of surface properties of calcium phosphate by means of photoacoustic spectroscopy (FT-IR/PAS), potentiometric titration and electrophoretic measurements The European Physical Journal Special Topics 154 (2008) 329-333  

Author Response

Thank you very much for your suggestions and for your time. We are very pleased to see the high appreciation of our manuscript. Our point-by-point responses are given below. The revisions in the manuscript are highlighted in green.

Question 1: Figures 4, 5 and 6 should be corrected.

Response: Thank you for the noticing. We have increased the font size in these figures as well as the resolution.

Question 2: The publication should describe the purity of the reagents used for the synthesis.

Response: The purity of initial reagents was specified in the section M&M (see page 2, line 83-84).

Question 3: It would be good to write what the Ca/P ratio was in individual samples after Mg substitution. 

Response: The Ca/P ratio depends on Mg concentration; the higher the dopant concentration, the lower the Ca/P ratio. Ca/P values have been added in Table 3.

Question 4: Novelty elements should be better highlighted in the introduction. Papers should be cited in Introduction section; for example:

The influence of the hydroxyapatite synthesis method on the electrochemical, surface and adsorption properties of hydroxyapatite Adsorption Science & Technology   35(5–6) (2017) 507–518

A study of surface properties of calcium phosphate by means of photoacoustic spectroscopy (FT-IR/PAS), potentiometric titration and electrophoretic measurements The European Physical Journal Special Topics 154 (2008) 329-333  

Response: Novelty elements are highlighted in the last paragraph of the Introduction.

Although the listed articles are not relevant to the topic of the current manuscript, we used one of them for citation because this article confirms the importance of the synthesis method. Thanks for this recommendation.

Round 2

Reviewer 1 Report

Comments and Suggestions for Authors

The manuscript ” Influence of magnesium source on the mechanochemical synthesis of magnesium-substituted hydroxyapatite” by  Bulina et al. described the preparation of Mg-substituted hydroxyapatite (Mg-HA) using the fast mechanochemical method with different magnesium sources.

The writing lacks clarity and sharpness, and is poorly organized in results and discussion section.  Therefore, I cannot recommend the acceptance of this work in the current form for publication in Materials.

Comments:

1.      Please explain the initial concentration of Mg (x) range from 0~2, why? 

2.      Please show the bone’s mechanical properties of Mg-substituted hydroxyapatite (Mg-HA) with an increase the initial concentration of Mg (x).

3.      Line 156, the significant digits of EHA should be shorted.

4.      The paper should be shortened to become more concise.

Author Response

Dear Reviewer,

Comments submitted in Round 2 contain the same questions as in Round 1.

Our detailed answers to these questions were already presented in the reviewer's report last time. Below we repeat them again.

Question 1: Please explain the initial concentration of Mg (x) range from 0~2, why?  

Response: The aim of the present work was to explore the possibility of Mg-HAs preparation by simple mechanochemical method using different magnesium sources, and to study the structure and thermal stability of the as-synthesized Mg-HAs. The range of the magnesium concentrations was chosen based on the literature data. The results of this study showed that the introduction of magnesium into the crystal lattice of HA reduces crystallinity. At x=2, a significant decrease in the crystallinity is observed, which indicates the existence of a substitution limit for the mechanochemical method of synthesis. In the present study, the limit was not reached, but unfortunately it is not possible to continue synthesis with higher x values at the moment. The discussion concerning the existence of a substitution limit for the mechanochemical method has been added to the manuscript.

Question 2: Please show the bone’s mechanical properties of Mg-substituted hydroxyapatite (Mg-HA) with an increase the initial concentration of Mg (x).

Response: Thank you for this recommendation. Unfortunately, such studies are impossible for this material due to its low thermal stability. Mechanical properties are commonly studied on a ceramic material annealed at high temperature. We have shown that the crystal lattice of Mg-substituted HA is destroyed at ~700 C, so it is impossible to make any ceramic product from this material without destroying the crystal structure of Mg-HA.

Question 3: Line 156, the significant digits of EHA should be shorted.

Response: Thanks for noticing. We have rounded the indicated value.

Question 4: The paper should be shortened to become more concise.

Response: In response to this comment, we critically analyzed the text of our manuscript and came to the conclusion that there is nothing to remove there. The section Result & Discussion contains three short subsections devoted to 1) mechanochemical synthesis; 2) modelling by DFT; 3) thermal stability of the material.  All information provided is important. Moreover, other Reviewers note “logical construction of article” and “good scientific rigor”.